# Transcriptional Profiling of the Effect of *Coleus amboinicus* L. Essential Oil against *Salmonella* Typhimurium Biofilm Formation

**DOI:** 10.3390/antibiotics12111598

**Published:** 2023-11-07

**Authors:** Arpron Leesombun, Sivapong Sungpradit, Ladawan Sariya, Jarupha Taowan, Sookruetai Boonmasawai

**Affiliations:** 1Department of Pre-Clinic and Applied Animal Science, Faculty of Veterinary Science, Mahidol University, Nakhon Pathom 73170, Thailand; arpron.lee@mahidol.edu (A.L.); sivapong.sun@mahidol.edu (S.S.); 2The Monitoring and Surveillance Center for Zoonotic Diseases in Wildlife and Exotic Animals (MoZWE), Faculty of Veterinary Science, Mahidol University, Nakhon Pathom 73170, Thailand; ladawan.sar@mahidol.edu (L.S.); namaoy.tao@mahidol.edu (J.T.)

**Keywords:** antibiofilm, essential oil, transcriptome, motility genes, curli fimbriae genes, invasion genes

## Abstract

*Salmonella enterica* serovar Typhimurium cause infections primarily through foodborne transmission and remains a significant public health concern. The biofilm formation of this bacteria also contributes to their multidrug-resistant nature. Essential oils from medicinal plants are considered potential alternatives to conventional antibiotics. Therefore, this study assessed the antimicrobial and antibiofilm activities of *Coleus amboinicus* essential oil (EO-CA) against *S.* Typhimurium ATCC 14028. Seventeen chemical compounds of EO-CA were identified, and carvacrol (38.26%) was found to be the main constituent. The minimum inhibitory concentration (MIC) of EO-CA for *S.* Typhimurium planktonic growth was 1024 µg/mL while the minimum bactericidal concentration was 1024 µg/mL. EO-CA at sub-MIC (≥1/16× MIC) exhibited antibiofilm activity against the prebiofilm formation of S. Typhimurium at 24 h. Furthermore, EO-CA (≥1/4× MIC) inhibited postbiofilm formation at 24 and 48 h (*p* < 0.05). Transcriptional profiling revealed that the EO-CA-treated group at 1/2× MIC had 375 differentially expressed genes (DEGs), 106 of which were upregulated and 269 were downregulated. Five significantly downregulated virulent DEGs responsible for motility (*flhD*, *fljB*, and *fimD*), curli fimbriae (*csgD*), and invasion (*hilA*) were screened via quantitative reverse transcription PCR (qRT-PCR). This study suggests the potential of EO-CA as an effective antimicrobial agent for combating planktonic and biofilm formation of *Salmonella*.

## 1. Introduction

Gastrointestinal infections occur when individuals consume food contaminated with enteric pathogens, such as *Salmonella* species, which are common causes of foodborne illnesses and significant contributors to diarrheal diseases. The United States Center for Disease Control and Prevention estimates that *Salmonella* bacteria cause ~1.35 million infections, leading to 26,500 hospitalizations and 420 deaths in the United States annually [1]. Food is the source of most of these illnesses. Several common serovars, such as *Salmonella enteritica* serovar Typhimurium and *S.* Enteritidis, are the main species responsible for causing human infections [2,3,4,5,6]. Animals, including livestock, poultry, pets, and wildlife, constitute the primary source of this pathogen, with animal-based foods serving as the main route of transmission to humans [7,8].

*Salmonella* can develop biofilm to provide resistance to stressful environments, including disinfectants, antibiotics, temperature fluctuations, and oxygen levels [2]. *Salmonella* biofilms are a major global public health concern because this characteristic holds particular significance for contamination in food processing and the food industry [9]. Biofilm formation is a multistep developmental process. First, bacteria attach to the carrier surface, which can either be abiotic (such as plastic, glass, rubber, cement, and stainless steel) or biotic (such as epithelial cells, gallstones, and plant surfaces). They form microcolonies and subsequently release extracellular polymeric substances, which include polysaccharides, including cellulose; proteins, such as curli, amyloid fimbria, and BapA protein; and DNA. These substances comprise the biofilm matrix, enhancing cell–surface adhesion and cell–cell interactions [10,11,12]. *S.* Typhimurium utilizes biofilms to achieve persistence in host and nonhost environments. Biofilms not only increase the virulence and persistent infection ability of *Salmonella* but also promote the development of antibiotic resistance, a crucial factor in the survival of *Salmonella* in challenging environments [13,14,15,16,17]. Therefore, biofilm inhibition can be potentially targeted by natural products or synthetic analogs [18]. Recently developed novel technologies, such as transcriptomics, are essential tools in the field of biological sciences. Transcriptomic analysis enables the comprehensive evaluation of differentially expressed genes (DEGs) and enrichment pathways in samples, rendering this a crucial method for understanding the genetic and biochemical mechanisms underlying antimicrobial resistance [19].

Plants have been considered a major source of medicines. They may help meet the urgent demand for alternatives to conventional antibiotics and can effectively inhibit and eradicate biofilms [20]. Essential oils can be extracted from oil-like volatile substances present in aromatic plant materials such as fruits, bark, seeds, pulp, peel, roots, and whole plants [21]. Several studies have reported the antisalmonella effects of essential oils extracted from *Lippia origanoides* [22], fresh leaves of *Ocimum gratissimum* [23], *Jatureja hortensis* [24], clove [25], and cinnamon [26]; however, few studies have assessed the effect of *Coleus amboinicus* essential oil (EO-CA) on *Salmonella* [27].

*C. amboinicus* Lour. (synonyms: *Plectranthus amboinicus* (Lour.) Spreng.), also known as Indian borage, belongs to the family Lamiaceae and is a perennial herb. This plant grows naturally and is distributed in tropical and subtropical regions. *C. amboinicus* exhibits a range of biological properties and is extensively used in traditional medicine to treat conditions such as fever, headaches, coughs, colds, asthma, constipation, and skin diseases [28]. *P. amboinicus* essential oil comprises volatile compounds belonging to various phytochemical classes, including terpenoids, phenolics, flavonoids, esters, alcohols, and aldehydes [28]. *P. amboinicus* essential oil reportedly exhibits pharmacological activity against various pathogens, including Gram-negative bacteria (*Escherichia coli*, *Pseudomonas aeruginosa*, and *S.* Typhimurium) and Gram-positive bacteria (*Staphylococcus aureus* and *Listeria monocytogenes*) [27], fungi [29], and insects [30]. Additionally, the essential oil of *C. amboinicus* has been assessed against a strain of *Streptococcus mutans* [31]. Several studies have reported the antibiofilm effects of the essential oil of *C. amboinicus* on pathogens such as oxacillin- and vancomycin-resistant *S. aureus* [32] and *Microsporum canis* [29]; however, studies regarding its effect on *S.* Typhimurium biofilm remain limited.

This study aimed to evaluate the chemical constituents of *C. amboinicus* essential oil (EO-CA) and assess its antimicrobial and antibiofilm activities against *S.* Typhimurium ATCC 14028. Furthermore, transcriptional profiling was conducted to unveil the overall patterns of gene expression, and the differential expression of genes involved in motility, curli fimbriae, and invasion was verified using quantitative reverse transcription PCR (qRT-PCR).

## 2. Results

### 2.1. Extraction and Composition of the Essential Oil

EO-CA was produced at a yield of 1.0% (*v*/*w*). The essential oil had a clear, light yellowish appearance and the oil properties at 20 °C were a refractive index of 1.498, density of 0.911 g/cm^3^, and specific gravity of 0.913. The chemical composition of EO-CA was determined via gas chromatography–mass spectrometry (GC–MS), and a total of 17 compounds were identified, representing 94.17% of the total composition (Table 1). The main compounds included carvacrol (38.26%), γ-terpinene (12.29%), and *p*-cymene (8.62%). The main phytochemical classes were monoterpenes (65.03%), phenols (0.56%), sesquiterpenes (10.89%), and fatty acids (17.71%).

The concentration of carvacrol was 4619.86 ± 217.84 µg/mL. Figure 1 displays a chromatogram of the primary constituents of EO-CA, while the mass spectrum is presented in Appendix A.

### 2.2. Determination of Antimicrobial and Antibiofilm Activities of EO-CA

The minimum inhibitory concentration (MIC) of EO-CA for *S.* Typhimurium as determined via microbroth dilution was 1024 µg/mL. The minimum bactericidal concentration (MBC) was 1024 µg/mL. Based on the MBC/MIC ratio, EO-CA exhibited a bactericidal effect against *S.* Typhimurium.

EO-CA at sub-MIC (1/2×, 1/4×, 1/8×, and 1/16× MIC) significantly inhibited prebiofilm formation with inhibition rates of 68.07% ± 1.09%, 47.34% ± 1.31%, 40.03% ± 5.52%, and 15% ± 1.97%, respectively (Figure 2). After *S.* Typhimurium had formed a biofilm at 24 h, bacteria were subjected to different EO-CA concentrations at 8×, 4×, 2×, 1×, and 1/2× MIC, resulting in significant inhibition of postbiofilm formation (*p* < 0.001). Exposure to EO-CA at 1/4× MIC also significantly inhibited postbiofilm formation (*p* < 0.05). After biofilm formation by *S.* Typhimurium at 48 h, bacteria were exposed to various EO-CA concentrations at 8×, 4×, 2×, 1×, 1/2×, and 1/4× MIC, leading to significant inhibition of postbiofilm formation (*p* < 0.001) (Figure 3).

### 2.3. Global Changes at the Transcriptome Level

Transcriptome sequencing was conducted using RNA extracted from the control and EO-CA (1/2× MIC)-treated groups. A total of 9.84 Gb of data were produced, including 103.14 raw reads and 98.33 clean reads. The two cDNA libraries for control and treated samples generated 55.17 and 43.16 clean reads, respectively. Table 2 provides a summary of the statistics related to the assembly of the transcriptome. The Q20 and Q30 percentages were ≥98% and 95%, respectively. Therefore, the subsequent analyses relied on high-quality data.

### 2.4. Analysis of DEGs

The inhibitory mechanism of EO-CA against *S.* Typhimurium was explored using RNA sequencing technology. DEG analysis included the following components: scatter plot, volcano map, heatmap, gene ontology (GO) analysis, and enrichment analysis of signaling pathways from the Kyoto Encyclopedia of Genes and Genomes (KEGG). A volcano map was used to visualize the overall distribution of the DEGs in both groups with a selection criterion of *p* < 0.05. In the EO-CA-treated group, there were 375 significant DEGs identified, comprising 106 upregulated and 269 downregulated genes (Figure 4 and Figure 5). The top three upregulated DEGs encoded putative arginine/ornithine antiporter, leucine-rich repeat protein, and acetolactate synthase, whereas the top three downregulated DEGs encoded glycerol-3-phosphate dehydrogenase, permeases of the drug/metabolite transporter superfamily, and autotransporter adhesin. A heatmap of the DEGs (FC > 2, false discovery rate [FDR] < 0.05) in the two groups is shown in Figure 6.

Based on the transcriptomic data, 911 genes were annotated into different GO terms. The most significantly enriched were biochemical processes (16 GO terms), cell components (8 GO terms), and molecular functions (8 GO terms) between the EO-CA-treated and control groups. Cell outer membrane (GO: 0009279), obsolete pathogenesis (GO: 0009405), and siderophore uptake transmembrane transporter activity (GO: 0015344) were strongly induced by EO-CA. Figure 7 displays the top GO terms induced during EO-CA adaptation.

Compared with the control group, 104 DEGs were enriched in 85 pathways in the group treated with EO-CA at 1/2× MIC, and the 21 most abundant pathways are indicated in Figure 8. The significantly enriched pathways included glycine, serine, and threonine metabolism; nitrogen metabolism; arginine biosynthesis; and bacterial chemotaxis.

The upregulated DEGs included genes that encode enzymes in the glycine, serine, and threonine pathways, including glycine C-acetyltransferase (EC 2.3.1.29) and glycerate 2-kinase (EC 2.7.1.165), whereas the downregulated DEGs included genes encoding phosphoglycerate mutase (EC 5.4.2.12) and glycine dehydrogenase (EC 1.4.4.2). DEGs were downregulated for genes encoding nitrogen pathway enzymes such as nitrate reductase (cytochrome) (EC 1.9.6.1), nitrite reductase (EC 1.7.1.15), nitrite reductase (cytochrome; ammonia-forming) (EC 1.7.2.2), and glutamate synthase (EC 1.4.1.13). The downregulated DEGs also included those encoding arginine biosynthesis pathway enzymes, including aspartate transaminase (EC 2.6.1.1), acetylornithine aminotransferase (EC 2.6.1.11), ornithine carbamoyltransferase (EC 2.1.3.3), argininosuccinate synthase (EC 6.3.4.5), and arginine deiminase (EC 3.5.3.6). DEGs were downregulated for genes encoding proteins in the bacterial chemotaxis pathway, which includes methyl-accepting chemotaxis protein (MCP) and dipeptide ATP-binding cassette (ABC) transporter periplasmic substrate-binding protein (DppA).

### 2.5. qRT-PCR Validation of Gene Expression

qRT-PCR was used to validate the effect of treatment with EO-CA (1/2× MIC) on the transcription of virulence genes involved in motility (*flhD*, *fljB*, and *fimD*), curli fimbriae (*adrA*, *csgD*), and invasion (*hilA*) in *S.* Typhimurium. The expression levels of *flhD*, *fljB*, *fimD*, and *hilA* correlated with the results of the transcriptomic analysis (Table 3). In addition, the expression levels of *flhD*, *fljB*, *fimD*, *csgD*, and *hilA*, but not that of *adrA*, significantly decreased (*p* < 0.05) following EO-CA treatment (Figure 9).

### 2.6. Field-Emission Scanning Electron Microscopy

The biofilm formation of *S.* Typhimurium was observed using field-emission scanning electron microscopy (FESEM; Figure 10). *S.* Typhimurium biofilm adhered to the round coverslips with appendages within the biofilm that promoted intercellular connections and facilitated attachment to the substrata (Figure 10a,b). Additional, pili-like fimbria structures and flagella were observed in the control group of *S.* Typhimurium. However, compared with the control group, the samples treated with 1/2× MIC EO-CA exhibited significant decreases in the adhesion rate (Figure 10c,d). Additionally, damaged bacteria characterized by shrinkage and corrugation were observed on the round coverslips treated with EO-CA (Figure 10d), whereas no damaged bacteria were observed in the control group (Figure 10a–c). Thus, EO-CA demonstrated a notable inhibitory impact on the formation of *S.* Typhimurium biofilm.

## 3. Discussion

This study demonstrated that EO-CA exhibits antibacterial and antibiofilm effects against *S.* Typhimurium and that the antibiofilm effects involve inhibiting the expression of genes responsible for motility and biofilm formation. EO-CA demonstrated a bactericidal effect and inhibited planktonic cell growth of *S.* Typhimurium with an MIC value of 1024 µg/mL. This result is similar to that reported in a previous study where the MIC and bactericidal concentrations of *P. amboinicus* essential oil against *S.* Typhimurium were 1 and 2 mg/mL, respectively [27]. The hot water extract of *P. amboinicus* leaves at 1000 µg/mL also reportedly inhibits the growth of *S.* Typhimurium by 55% [33].

EO-CA initially inhibited *Salmonella* biofilm formation during the early stages of biofilm development at concentrations of ≥1/16× MIC (64 µg/mL). Furthermore, EO-CA inhibited postbiofilm formation after allowing *S.* Typhimurium biofilm development for 24 and 48 h. Importantly, significant inhibition was observed at the relatively low concentration of 1/4× MIC (256 µg/mL) in both experiments.

The yield of CO-EA with extraction through steam distillation was 1.0% (*v*/*w*), and the 17 compounds identified in EO-CA represented 94.17% of the total essential oil, with carvacrol being the main component at 38.26%. The essential oil of *P. amboinicus* exhibits significant differences in yield and composition that depend on factors such as the geographical area from where the plant is obtained, season, and extraction method [28]. Several correlations with climatic parameters have been observed, indicating that plants collected during the rainy season yield more oil extract [34]. Herein, carvacrol was the major component, which agrees with the results of previous studies that reported carvacrol as the major constituent, comprising 56.65%–88.17% [29,30,32,35,36].

Carvacrol exhibits potential inhibitory effects against the growth of *S.* Typhimurium, with an MIC and MBC of 312 µg/mL. Additionally, carvacrol at 2× and 4× MIC was effective against preformed biofilms on polypropylene and stainless steel surfaces [37]. Olsen et al. (2013) observed that when carvacrol is used at a sublethal concentration of 0.47 mM, it significantly downregulates the expression of key virulence genes of *Salmonella*, including *hilA*, *invA*, *prgH*, *sipA*, *sipC*, *sipD*, *sopB*, and *sopE2.* These genes play crucial roles in invasion and epithelial damage [38]. Several studies successfully used carvacrol as a substance to inhibit *S.* Typhimurium biofilms [39] and *Chromobacterium violaceum* quorum sensing (QS) [40]. Hence, the antibacterial and antibiofilm activities of EO-CA may be attributed to the presence of carvacrol in this essential oil.

Our transcriptome study identified the downregulated expression levels of genes involved in chemotaxis (GO: 0006935), signal transduction (GO: 0007165), signal complex assembly (GO: 0007172), detection of chemical stimulus (GO: 0009593), regulation of protein histidine kinase activity (GO: 0032110), receptor clustering (GO: 0043113), cell motility (GO: 0048870), regulation of chemotaxis (GO: 0050920), cellular response to amino acid stimulus (GO: 0071230), and regulation of bacterial-type flagellum-dependent cell motility (GO: 1902021). Additionally, the expression levels of motility genes (e.g., *flhD*, *fljB*, and *fimD*), curli fimbriae genes (e.g., *csgD*) and invasion genes (e.g., *hilA*) were significantly decreased when validated via qRT-PCR.

QS is involved in multiple cellular processes and particularly in bacterial virulence during biofilm formation, motility, adherence, and invasion. Strong evidence suggests that QS plays a critical role in bacterial biofilm formation [10]. QS serves as an intercellular communication system that enables bacteria to coordinate their population density and regulate a diverse range of physiological processes [13].

Several possible mechanisms involving plant metabolites could inhibit *Salmonella* biofilm formation through gene regulation involving QS. Kim et al. demonstrated that quercetin is an antioxidant that disrupts biofilm formation by suppressing the expression of genes associated with quorum sensing (*luxS*), virulence (*avrA* and *hilA*), and stress response (*rpoS*) [41]. The antimicrobial and antibiofilm effects of the essential oil from *L. origanoides*, which contains thymol (32.7%), carvacrol (18.8%), trans-β-caryophyllene (6.2%), and γ-terpinene (5.4%), were investigated during the formation of biofilm and expression of QS genes in *S.* Enteritidis 13076 and *S.* Typhimurium 14028. *L. origanoides* essential oil could inhibit the QS mechanism involving *luxR*, *luxS*, *QseB*, and *sdiA* as well as biofilm formation associated with *csgA*, *csgB*, *csgD*, *flhD*, *fliZ*, and *motB* [22]. Herein, transcriptome analysis revealed that *Salmonella* treated with EO-CA exhibited downregulated expression of genes associated with QS, including *luxR*, *luxS*, and *QseB* (fold change = −0.8157, −0.6612, and −0.4578, respectively). However, no significant differences in expression were observed between the treated and control groups.

Several essential oils have demonstrated inhibitory effects due to antibacterial and antibiofilm activities, as indicated via transcriptome analysis. Thyme and cinnamon oils affected *S.* Typhimurium biofilm formation while transcriptome analysis identified 161 and 324 DEGs, respectively. Cell motility and transmembrane material exchange were inhibited in the thyme-treated sample; however, they were activated in a cinnamon-treated sample [42]. The essential oils of clove (CEO) and oregano (OEO) effectively inhibited *Salmonella* Derby biofilm formation at 1/2× and 1/4× MIC (MICs of CEO = 0.8 mg/mL and OEO = 0.2 mg/mL). CEO and OEO significantly downregulated the expression of genes associated with energy metabolism pathways, particularly those involved in fatty acid degradation, oxidative phosphorylation, and tricarboxylic acid cycle pathways. Consequently, CEO and OEO inhibit the biofilm formation of *S.* Derby by suppressing its energy metabolism and impeding cellular activity [43]. Herein, we identified 375 DEGs in EO-CA-treated *S.* Typhimurium, 106 of which were upregulated and 269 were downregulated.

Flagella, which are the main motility organs of bacteria, contribute to processes such as chemotaxis, attachment to and invasion of host cells, and colonization, and even trigger innate immune responses during the biofilm formation process [44]. Additionally, flagella play important roles in biofilm initiation and maturation [45,46]. The major regulators of flagellar assembly are FlgM, FlhC, FlhD, FliA, FliD, FliT, and FliZ [47]. Flagella mutants (*ΔflgE* and *ΔfliC*) lack flagellar motility and exhibit lower biofilm formation during the early stage compared with wild-type cells, suggesting that flagellar motility plays a significant role in initial cell–surface interactions [45]. Herein, EO-CA affected *S.* Typhimurium by decreasing the expression of motility genes including *flhD* and *fljB* (flagellar biosynthesis regulator) and *fimD* (motility). *flhD* encodes the flagellar switch protein FlhD4C2, which functions as the central regulator and transcriptional activator of flagellar genes [48]. *fljB* encodes the phase II flagellin protein, which acts as a significant virulence factor, potentially playing a role in regulating virulence and pathogenesis [44,49]. The downregulation of genes involved in flagellar assembly, such as *fljB*, could inhibit cell motility [38]. *fimD* partially encodes type 1 fimbriae, which can contribute to functions such as adherence, invasion, biofilm formation, and immune response [50]. Thus, EO-CA affects the flagellar motility of *S.* Typhimurium by decreasing the expression of *flhD*, *fljB*, and *fimD*, and the decrease in bacterial motility may affect the early stage of bacterial biofilm formation.

The *csgD*, *csgA*, *adrA*, and *bcsA* genes are directly associated with biofilm components. In *S.* Enteritidis mutants, *ΔcsgD*, *ΔcsgA*, and *ΔbcsA* mutations impaired biofilm formation compared with that in *S.* Enteritidis wild-type strain C50041 [51]. *csgD* and *adrA* are involved in regulating biofilm formation and act as the primary transcriptional controller of biofilms by coordinating the regulation of curli, cellulose, and other polymers [11,52]. *csgD*, which is a FixJ/LuxR/UhpA family transcription factor, encodes a transcriptional regulator CsgD protein (Curlin subunit gene D) and regulates the *csgBAC* and *csgDEFG* operons responsible for curli component synthesis, secretion, and assembly. The CsgD protein additionally triggers the expression of *adrA* (a diguanylate cyclase), a crucial element in the synthesis of cellulose required for the formation of biofilms [52,53]. A difference in the expression of *adrA* was noted between the qRT-PCR and RNA sequencing results, and these differences could potentially arise from variances in sensitivity and specificity inherent to these two methodologies [54]. The decreased expression of genes associated with motility and curli fimbriae in this study appeared to inhibit biofilm formation. This observation was supported by FESEM images that showed fewer bacteria in the EO-CA-treated group as well as reduced pili-like fimbriae structures and flagella compared with the control group.

In addition, pathway analyses identified bacterial chemotaxis as the most abundant pathway that was affected by the downregulation of MCP. The MCP protein is responsible for the movement of *S.* Typhimurium through the mucosal epithelium [55] and is regulated by the regulatory protein HilD, which also regulates *hilA* in *Salmonella* pathogenicity island 1 [56]. Herein, the expression of *hilA* was downregulated, as revealed via both qRT-PCR and RNA sequencing analyses (fold change = −1.529).

The expression levels of *DppA*, which encodes the dipeptide ABC transporter periplasmic substrate-binding protein, were also downregulated in this study. Chaudhari et al. demonstrated membrane damage and formation of ghost cells in *S.* Typhimurium when exposed to silver and pegylated silver-coated single-walled carbon nanotubes, which were associated with a significant downregulation of *DppA* expression levels [57]. Damaged bacteria exhibiting shrinkage were also present on the round coverslips treated with EO-CA in this study. Therefore, the downregulated expression of *DppA* may be the consequence of EO-CA exposure that affected the bacterial cellular membrane and external structures.

## 4. Materials and Methods

### 4.1. Plant Material

*Coleus amboinicus* (Lour.) plant material was sourced in June 2022 from a pesticide-free garden located in Nonthaburi Province, central Thailand. The plant was identified and authenticated and a voucher specimen of the plant was deposited at Sireeruckhachati Nature Learning Park, Faculty of Pharmacy, Mahidol University (PBM No.005507-8). Steam distillation was used to extract the essential oil from 100 kg of freshly harvested *C. amboinicus* leaves. The obtained essential oil was stored in amber glass containers at 4 °C. The essential oil yield was calculated as a percentage of the fresh plant material’s weight and expressed as % (*v*/*w*). The density was determined using a density meter (DA-100M, Tokyo, Japan), and the refractive index was computed via a refractometer (RX-5000CX, Atago, Tokyo, Japan).

The chemical components of EO-CA were analyzed using a GC–MS instrument (model 7890A-MS5975C, Agilent Technologies, Santa Clara, CA, USA) equipped with a DB-5HT capillary column (30 m × 250 μm × 0.1 μm) (Agilent Technologies, USA). The sample was introduced using split mode with a split ratio of 1:10. Helium served as the carrier gas at a flow rate of 1 mL/min. The injection port maintained a temperature of 250 °C, and the column temperature program comprised an initial period at 40 °C for 5 min, followed by a gradual increase to 250 °C at a flow rate of 10 °C/min, and then held at 250 °C for an additional 5 min. For the mass spectrometry analysis, the conditions encompassed an ion source temperature of 250 °C, ionization energy of 70 eV, and mass scan range spanning from 30 to 550 m/z. Compounds were identified and authenticated by comparing their mass spectra to W10N14.L (Wiley 10th and NIST 2014 libraries).

### 4.2. Bacterial Strain and Culture

The study protocol was approved by the Faculty of Veterinary Science-Institutional Biosafety Committee (IBC/MUVS-B-002/2565). *S.* Typhimurium ATCC 14028 strain was cultured on 5% sheep blood agar and incubated at 37 °C.

### 4.3. Antimicrobial Activity of EO-CA

The MIC of EO-CA against *S.* Typhimurium was established using the broth dilution method, following the recommendations outlined by the Clinical and Laboratory Standards Institute (CLSI 2020) [58]. EO-CA was dissolved in Mueller–Hinton broth and then 100 µL of solution was added per well in a 96-well plate to final concentrations of 64, 128, 256, 512, 1024, 2048, and 4096 µg/mL. A single colony of *S.* Typhimurium was suspended in 0.85% NaCl, and the opacity was adjusted to 0.5 MacFarland (bacterial concentration ~1 × 10^8^ colony forming units (CFU)/mL). Then, 100 µL bacterial suspension was introduced into wells that already contained the essential oil treatments, resulting in a final bacterial concentration of 1 × 10^5^ CFU/mL. Ciprofloxacin was used as a positive control in the experiments. Plates were incubated for 18–24 h at 37 °C.

The MIC of the samples was assessed by adding 30 µL resazurin sodium salt per well, which was prepared at 0.02% (*w*/*v*) in sterile distilled water [59]. The samples were incubated for an additional 2–4 h. Viable bacteria reduced the blue dye (indicating no growth) to a purple or pink color. The MIC was defined as the minimum concentration of the essential oil at which there was no observable change in color.

To determine the MBC, 10 μL bacterial solution from each well that exhibited the blue color was plated on Mueller–Hinton agar and cultured for 24 h. The MBC of essential oil treatment was defined as the minimum concentration of essential oil that killed 99.9% of test bacteria.

### 4.4. Antibiofilm Formation of EO-CA

The potential of EO-CA to prevent initial bacterial cell attachment (prebiofilm formation) was investigated through the biofilm inhibition assay [60]. Bacteria were suspended in tryptic soy broth (TSB; Clinical Diagnostics Ltd., Bangkok, Thailand) and adjusted to 1 × 10^6^ CFU/mL. A volume of 100 µL suspension was introduced into sterile 96-well flat-bottom polystyrene microplates, each containing sub–inhibitory concentrations (sub-MIC) of EO-CA: 1/2×, 1/4×, 1/8×, and 1/16× MIC. The microplates were then incubated at 37 °C for 24 h without any shaking, and the biomass of the biofilm was assessed using crystal violet staining [61].

To assess the reduction in biofilm mass following biofilm formation (postbiofilm formation), bacteria were incubated at 37 °C for 24 and 48 h to establish biofilm. Then, the supernatant was removed, and each well was washed thrice with 200 µL sterile phosphate-buffered saline (PBS; pH 7.4). Fresh medium containing EO-CA at concentrations of 8×, 4×, 2×, 1×, 1/2×, and 1/4× MIC was added at 200 µL per well. The plates were further incubated at 37 °C for 24 h, and subsequently, the biofilm biomass was quantified using crystal violet staining.

### 4.5. Crystal Violet Staining

The supernatant was removed and then each well was gently rinsed thrice with 200 µL of PBS. The remaining attached bacteria were fixed with 200 µL 95% methanol and subsequently stained with 150 µL 0.3% crystal violet for 15 min at room temperature (RT). Excess dye was eliminated by rinsing with PBS. After allowing the plates to air-dry, the biofilms that had been stained were redissolved in 150 µL 33% (*v*/*v*) glacial acetic acid. The biofilm biomass was quantified via calculating the optical density at 590 nm using a micro-ELISA automatic plate reader (BIOTEK, Winooski, VT, USA). Each experiment was performed thrice.

### 4.6. FESEM

FESEM was employed to observe the effects of EO-CA on *S.* Typhimurium. Sterile round coverslips (18-mm diameter; NUNC™, New York, NY, USA) were dropped onto a 12-well plate. Each well containing EO-CA at sub-MIC (1/2× MIC; 512 µg/mL) was inoculated with *S.* Typhimurium at 5 × 10^5^ CFU/mL, and the treated plates were then incubated for 24 h at 37 °C. The supernatant was removed, the round coverslips were washed thrice with PBS and fixed in 2.5% glutaraldehyde (Sigma-Aldrich, St. Louis, MO, USA) for 60 min, and the attached cells and biofilms were dehydrated in an ethanol gradient (20%, 40%, 60%, 80%, and 100%) for 10 min at RT [22,62]. The round coverslips were dried and mounted onto stubs using double-sided carbon tape. Each sample was coated with a thin layer of platinum using a sputter coater (JEC-3000FC, JEOL Ltd., Tokyo, Japan) and examined using FESEM (JSM-7610FPlus, JEOL Ltd., Tokyo, Japan) at 10 kV.

### 4.7. RNA Extraction

*S.* Typhimurium was cultured in TSB with and without EO-CA at 1/2× MIC (512 µg/mL) as the treated and control groups, respectively. Bacteria were incubated at 37 °C for 24 h. *S.* Typhimurium culture (1 mL) was centrifuged at 5000× *g* for 10 min. The bacterial pellet was resuspended with Tris-EDTA buffer (10 mM Tris-HCl, 1 mM EDTA, pH 8.0) containing 15 mg/mL lysosome and incubated at RT for 10 min. Bacterial cells were lysed using RLT buffer combined with mechanical disruption using glass beads. Total RNA was subsequently extracted using an RNeasy Mini kit (QIAGEN, Hilden, Germany) alongside DNase digestion. A spectrophotometer (Nanodrop One, Thermo Fisher Scientific, Waltham, MA, USA) was used to measure the quality and concentration of RNA in the samples. The integrity of RNA was assessed through 1% (*w*/*v*) agarose gel electrophoresis, and the RNA samples were stored at −80 °C.

### 4.8. Transcriptomics Analysis

RNA sequencing was performed to investigate the responses of *S.* Typhimurium at sub-MIC EO-CA (1/2× MIC). Three biological replicates in each group of RNA extraction were pooled and processed without technical replicates. Following RNA extraction, pure RNA samples without contamination were further enriched using oligo(dT) beads. The RNA Integrity Number was assessed via an Agilent 2100 Bioanalyzer system (Agilent Technology, USA). The obtained mRNA was randomly fragmented by adding fragmentation buffer, and cDNA was synthesized using dUTP. The library was amplified to create DNA nanoballs and sequenced on the DNBSEQ platform (BGI Genomics, Shenzhen, China).

Data filtering was performed using Bowtie2 (version: 2.4.5) to map the clean reads to the ribosomal database, remove the reads mapped to the ribosomal database, and use the retained data for subsequent analysis. Following the filtration process, the remaining reads were labeled as “clean reads” and archived in FASTQ format for subsequent bioinformatics analysis. The filtered data (clean reads) were used for transcript assembly. Trinity (version: v2.13.2) was used for assembly, and cdhit (version: 4.6.1) was used to remove redundancy from the assembled transcripts. Finally, the quality of the assembled transcripts was evaluated using the single-copy direct homologous database BUSCO (Version: v5.4.3). According to the assembly results, the single sequence repeats of the transcript were detected and primers were designed. After obtaining the new transcripts, the coding ability was predicted using three predictive software tools lncRNA (CPC, LGC-1.0, and CNCI) and two protein databases (Pfam and SwissProt) to distinguish between mRNA. For quantitative analysis, the Boetie2 and RSEM (version v1.3.1) programs were used to calculate gene and transcript expression.

Differential analysis involved comparing the expression values of samples, excluding genes with subtle changes in expression, and retaining those with significant differences. This comparison was conducted between two samples (typically, control and treatment) using PoissonDis software [63] for differential analysis. A volcano plot was employed to evaluate the general distribution of genes that were differentially expressed.

To identify genes with similar expression trends, pheatmap (version 1.0.12) was used to cluster the expression of genes. BLAST (version 2.7.1) was used to align mRNA to NT; Diamond was used to align mRNA to NR, KOG, KEGG, UniProt, COG, and TF (animal transcription factor database, AnimalTFDB3; plant transcription factor database, PlantTFdb5) databases. GO annotation was based on the alignment information of UniProt (including sprot and trembl). Infernal (version 1.1.4) was used to align lncRNA/mRNA to Rfam; Diamond (version 2.0.15.153) was used to align lncRNA to miRBase (miRNA precursor sequences).

GO and KEGG are the two most functional databases used for gene classification. To quantitatively assess the enrichment of candidate genes within specific functional modules, a hypergeometric model was used to calculate the enrichment for each functional module. Each functional module corresponded to a *p*-value. The smaller the *p*-value, the more abundant the gene in a functional module. FDR correction was subsequently performed based on the *p*-value; an FDR of ≤0.01 was set for significant enrichment.

### 4.9. Gene Expression Analysis

To investigate the impact of EO-CA on the transcription of genes associated with biofilm formation, genes were selected based on the RNA sequencing results and a previous study [64]. All primer sequences, covering the forward and reverse sequences, can be found in Table 4. The expression of *S.* Typhimurium genes treated with EO-CA at 1/2× MIC was assessed using qRT-PCR based on SYBR^®^ Green I fluorescence. cDNA was synthesized from total RNA using the SuperScript^TM^ III First-Strand Synthesis System (Thermo Fisher Scientific, Waltham, MA, USA). Then, the qPCR reaction was conducted with a mixture of 10 ng cDNA, 10 μL QuantiNova SYBR^®^ Green PCR master mix (QIAGEN, Hilden, Germany), and 0.2 µM forward and reverse primers using an QuantStudio^TM^ 3 Real-Time PCR System (Thermo Fisher Scientific, Waltham, MA, USA), with an annealing temperature of 60 °C for all the targeted genes and 62 °C for the reference gene, *16S rRNA*. Data were analyzed using the 2^−(ΔΔCT)^ method to calculate the relative expression (Log10 fold change) of the target genes of the *S.* Typhimurium treatment group after normalization with the *16S rRNA* gene relative to the *S.* Typhimurium control group.

### 4.10. Statistical Analysis

Data are presented as the mean ± standard deviation. The significance of differences between the control and treatment values was determined using one-way analysis of variance. All statistical analyses were performed using GraphPad Prism 6 (GraphPad Software, Inc., La Jolla, CA, USA). In all cases, a *p*-value of <0.05 was considered significant.

## 5. Conclusions

EO-CA demonstrated bactericidal properties against *S.* Typhimurium at a concentration of 1024 µg/mL and inhibited both prebiofilm and postbiofilm formation at ≥1/4× MIC. The primary constituent of EO-CA, carvacrol, is considered a leading candidate among plant-based compounds with potential to inhibit bacterial growth. This study revealed the molecular mechanism underlying the antibiofilm activity of EO-CA at 1/2× MIC via the downregulated expression of genes involved with motility (*flhD*, *fljB*, *fimD*) and curli fimbriae (*csgD*). In addition, EO-CA suppressed the expression of the invasion gene (*hilA*). Thus, the findings of this study strongly suggest the potential of EO-CA as an effective antimicrobial agent to combat *Salmonella* planktonic growth and biofilm formation.

## Figures and Tables

**Figure 1 antibiotics-12-01598-f001:**
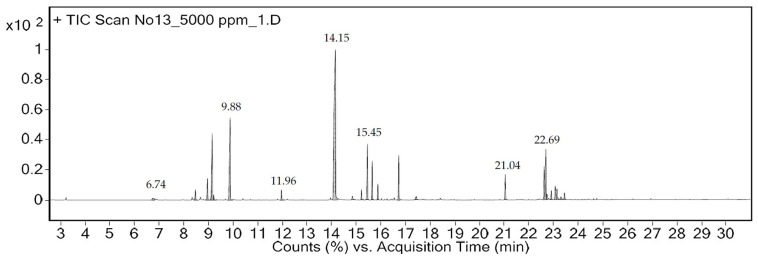
Chromatogram generated to determine the primary components of *Coleus amboinicus* essential oil using gas chromatography–mass spectrometry.

**Figure 2 antibiotics-12-01598-f002:**
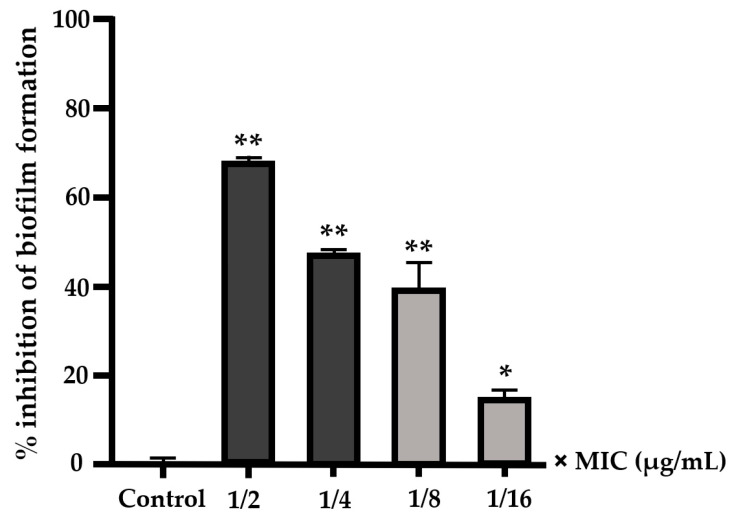
Percentage of inhibition of prebiofilm formation by *S.* Typhimurium. Bacteria were treated with EO-CA at 1/2×, 1/4×, 1/8×, and 1/16× MIC for 24 h. Data are presented as mean ± SD. ANOVA was employed to demonstrate significant differences compared to the control, where * indicates *p* < 0.05, and ** indicates *p* < 0.001.

**Figure 3 antibiotics-12-01598-f003:**
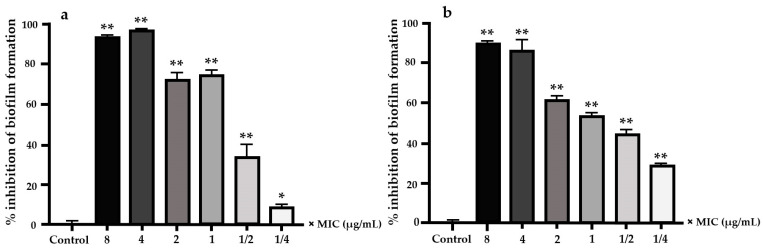
Percentage of inhibition of postbiofilm formation by *S.* Typhimurium. Bacteria were treated with EO-CA at 8×, 4×, 2×, 1/2×, and 1/4× MIC after producing biofilm for (**a**) 24 h or (**b**) 48 h. Data are presented as mean ± SD. ANOVA was employed to demonstrate significant differences compared to the control, where * indicates *p* < 0.05, and ** indicates *p* < 0.001.

**Figure 4 antibiotics-12-01598-f004:**
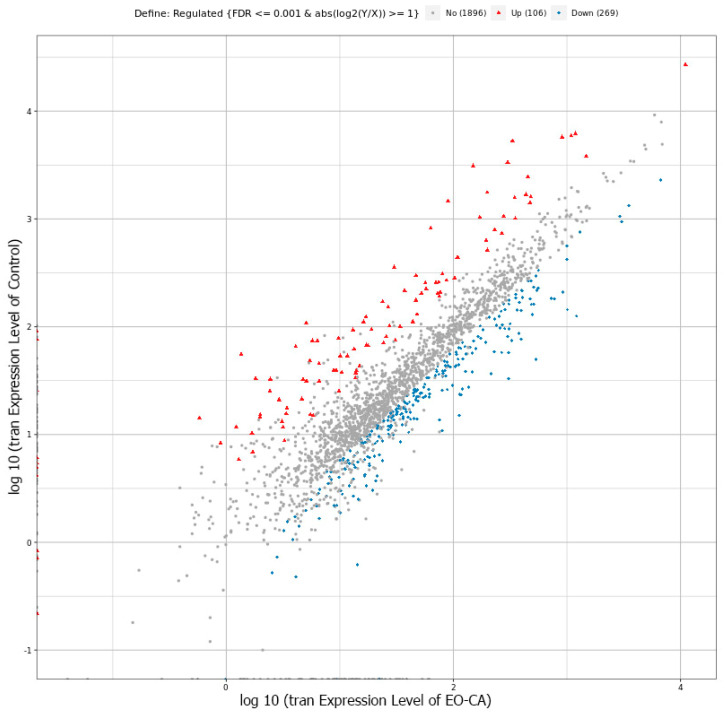
Scatter plot of the transcription levels of significantly upregulated and downregulated DEGs in the control group versus the EO-CA-treated group. The figure shows the distribution of expression levels. *X*-axis: the logarithmic value of gene expression of the EO-CA-treated group; *Y*-axis: the logarithmic value of gene expression of the control group in comparison; Colors: blue indicates the downregulated genes; orange indicates the upregulated genes; brown indicates nonsignificant DEGs.

**Figure 5 antibiotics-12-01598-f005:**
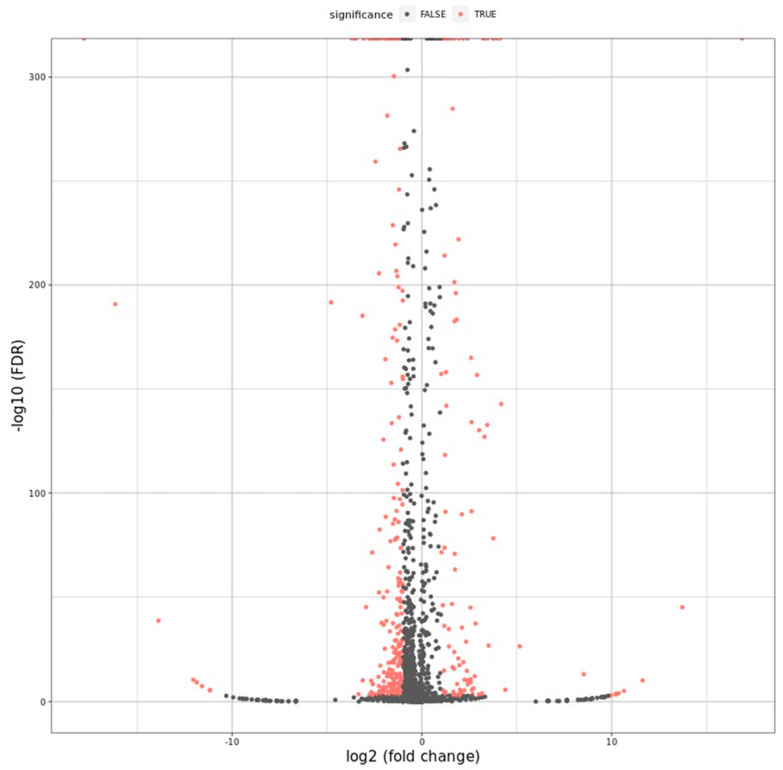
Volcano map of transcription levels in the control group versus the EO-CA-treated group. The *X*-axis shows the log2-transformed fold change, while the *Y*-axis shows the log10-transformed significance value. Colors: red indicates the significant DEGs; gray indicates the nonsignificant DEGs.

**Figure 6 antibiotics-12-01598-f006:**
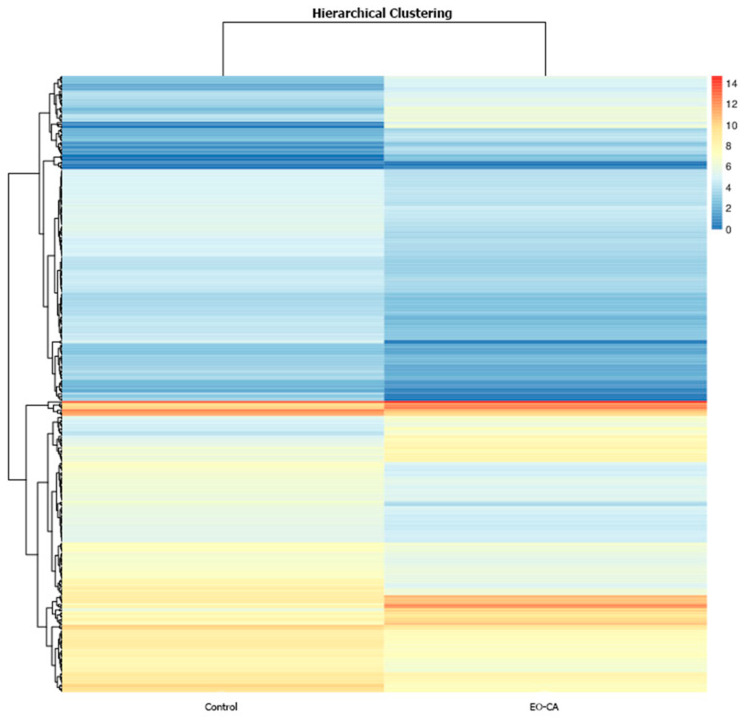
Heatmap illustrating DEGs in the control group versus the EO-CA-treated group. *Y*-axis: RNA; color: cluster value.

**Figure 7 antibiotics-12-01598-f007:**
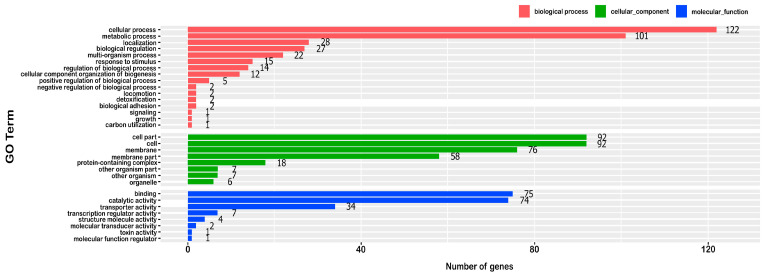
GO term enrichment analysis of the significant transcript levels in the control group versus the EO-CA-treated group. *X*-axis: number of genes; *Y*-axis: GO term; Color: class name of top level.

**Figure 8 antibiotics-12-01598-f008:**
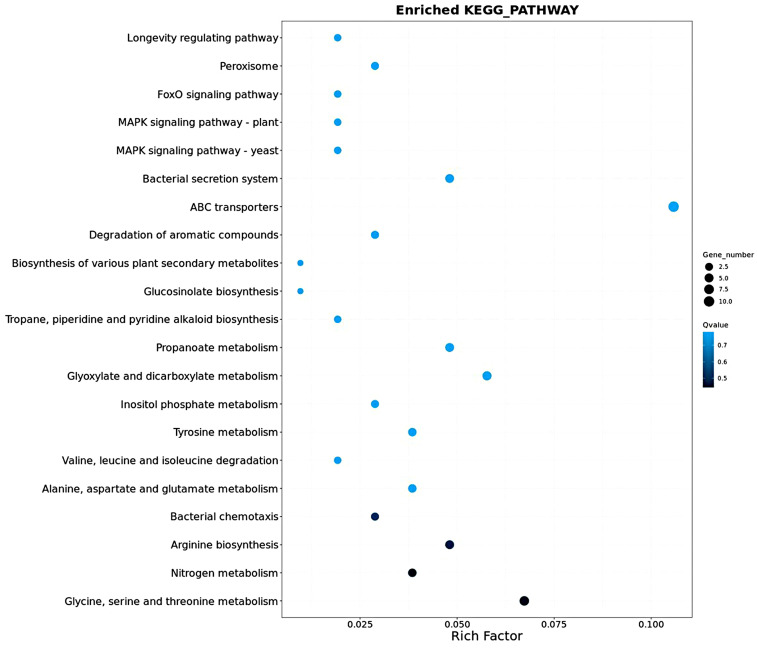
Most abundant pathways for a total of 104 DEGs. The dot size corresponds to the number of genes, and the degree of enrichment increases as the *q*-value approaches 0.5.

**Figure 9 antibiotics-12-01598-f009:**
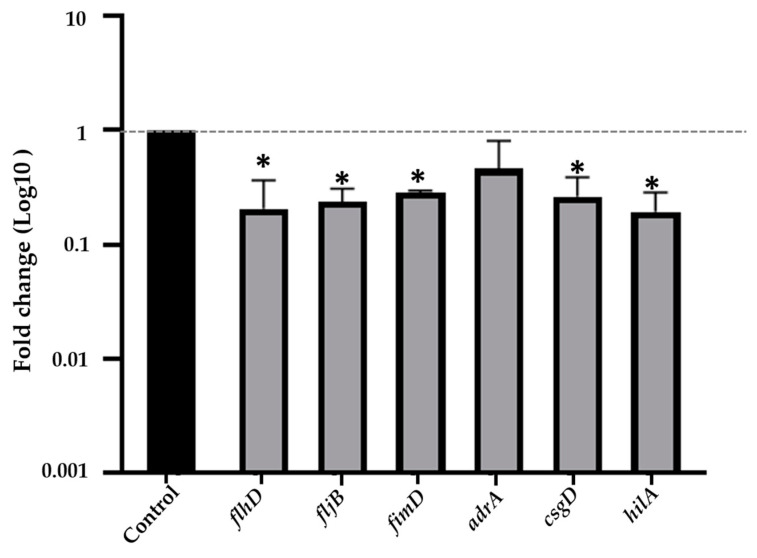
Gene expression patterns of virulence-related genes in biofilm-associated cells. Effect of EO-CA (1/2× MIC) on *S.* Typhimurium gene expression. The expression levels of the target genes were normalized using the reference gene *16S rRNA*. The bars represent the standard deviation. ANOVA was employed to demonstrate significant differences compared to the control, where * indicates *p* < 0.05.

**Figure 10 antibiotics-12-01598-f010:**
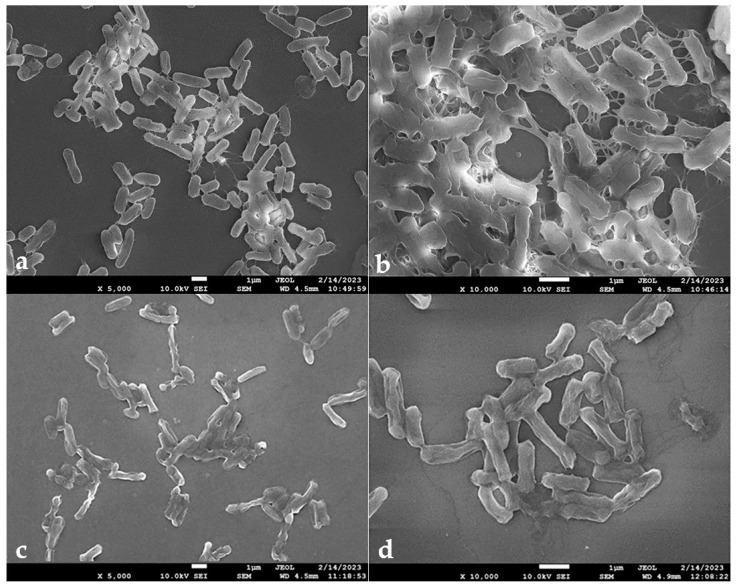
Antibiofilm activity of EO-CA against *S.* Typhimurium as observed by FESEM. (**a**,**b**) Biofilm growth on round coverslips after 24 h of incubation. (**c**,**d**) Biofilm growth on round coverslips after treatment with EO-CA at 1/2× MIC after 24 h of incubation (magnification of panels from the left: 5000×, 10,000×).

**Table 1 antibiotics-12-01598-t001:** Chemical composition of *Coleus amboinicus* essential oil.

No	Retention Time	Classes	Compounds	Formula	ChemicalStructure	% of Total
1	6.74	Monoterpene	α-Thujene	C_10_H_16_	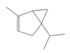	0.60
2	8.47	Monoterpene	β-Myrcene	C_10_H_16_	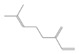	1.15
3	8.96	Monoterpene	α-Terpinene	C_10_H_16_	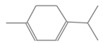	2.49
4	9.16	Monoterpene	*p*-Cymene	C_10_H_14_	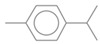	8.62
5	9.21	Monoterpene	Isoterpinolene	C_10_H_16_	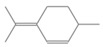	0.58
6	9.88	Monoterpene	γ-Terpinene	C_10_H_16_	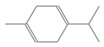	12.29
7	11.96	Monoterpene	(−)-Terpinen-4-ol	C_10_H_18_O	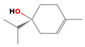	1.04
8	14.15	Monoterpene	Carvacrol	C_10_H_14_O	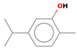	38.26
9	14.84	Phenols	Eugenol	C_10_H_12_O_2_	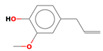	0.56
10	15.45	Sesquiterpene	Caryophyllene	C_15_H_24_	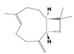	5.64
11	15.64	Sesquiterpene	*cis*-α-Bisabolene	C_15_H_24_	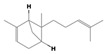	3.71
12	15.88	Sesquiterpene	Humulene	C_15_H_24_	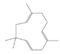	1.54
13	16.72	Fatty acids	Dodecanoic acid, methyl ester	C_13_H_26_O_2_	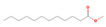	4.35
14	17.43	Fatty acids	Dodecanoic acid	C_12_H_24_O_2_	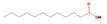	0.73
15	21.04	Fatty acids	Hexadecanoic acid, methyl ester	C_17_H_34_O_2_	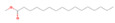	2.27
16	22.62	Fatty acids	Linoleic acid	C_19_H_34_O_2_	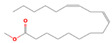	3.24
17	22.69	Fatty acids	9-Octadecenoic acid, methyl ester	C_19_H_36_O_2_	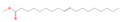	7.12
		Total				94.17

**Table 2 antibiotics-12-01598-t002:** Summary of RNA sequencing alignment.

Sample Name	Total Raw Reads (Mb)	Total Clean Reads (Mb)	Total Clean Bases (Gb)	Clean Read Q20 (%)	Clean Read Q30 (%)	Clean Read Ratio (%)	GC (%)
Control	57.94	55.17	5.52	98.55	94.94	95.22	52.17
EO-CA	45.20	43.16	4.32	98.65	95.27	95.49	52.21

EO-CA: *C. amboinicus* essential oil; Q20: percentage of bases with a Phred value >20; Q30: percentage of bases with a Phred value >30.

**Table 3 antibiotics-12-01598-t003:** DEG expression data.

Gene Name	RNA-Seg Significant Fold Change (Control vs. EO-CA)	Significant Different Value	Corrected Significant Different Value	Significant (Control vs. EO-CA)
*flhD*	−1.6865	1.51 × 10^−13^	5.16 × 10^−13^	Downregulated
*fljB*	−1.5424	2.01 × 10^−176^	2.37 × 10^−175^	Downregulated
*fimD*	−1.9259	4.38 × 10^−166^	4.95 × 10^−165^	Downregulated
*adrA*	−1.4871	2.37 × 10^−155^	2.18 × 10^−114^	Downregulated
*csgD*	−0.6398	1.50 × 10^−7^	4.22 × 10^−7^	Not significant
*hilA*	−1.5295	6.03 × 10^−39^	3.28 × 10^−38^	Downregulated

EO-CA; *Coleus amboinicus* essential oil.

**Table 4 antibiotics-12-01598-t004:** Primer sequences.

Gene	Gene Function	Forward (5′−3′)	Reverse (5′−3′)
*16S rRNA*		AGGCCTTCGGGTTGTAAAGT	GTTAGCCGGTGCTTCTTCTG
*flhD*	Motility	CTCCTTGCACAGCGTTTGAT	TCTCCGCCAGTTTGACCAT
*fljB*	Motility	TGGATGTATCGGGTCTTGATG	CACCAGTAAAGCCACCAATAG
*fimD*	Motility	CGCGGCGAAAGTTATTTCAA	CCACGGACGCGGTATCC
*adrA*	Curli fimbriae	GAAGCTCGTCGCTGGAAGTC	TTCCGCTTAATTTAATGGCCG
*csgD*	Curli fimbriae	TCCTGGTCTTCAGTAGCGTAA	TATGATGGAAGCGGATAAGAA
*hilA*	Invasion	AATGGTCACAGGCTGAGGTG	ACATCGTCGCGACTTGTGAA

## Data Availability

The data presented in this study are available within the article.

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
