# Peer review of "Transcriptional Profiling of the Effect of Coleus amboinicus L. Essential Oil against Salmonella Typhimurium Biofilm Formation"

_antibiotics, 2023, doi:10.3390/antibiotics12111598_

Round 1
Reviewer 1 Report
Comments and Suggestions for Authors
In the submitted manuscript (ID: antibiotics-2689683), the authors convincingly demonstrated the antibacterial and antibiofilm inhibitory potential of Coleus amboinicus L. essential oil against Salmonella Typhimurium. They identified which compounds could have these important bioactive properties and transcriptionally profiled (possible) molecular targets of their action.
The general impression of the reviewed work is excellent; congratulations to the authors. I would especially emphasize and praise the study's design, the experimental protocols described in detail, and the language in which the manuscript was written. Therefore, the authors should continue researching this critical scientific field of plant-derived antimicrobial substances). My remarks and suggestions are minor and listed in the following two paragraphs.
The inhibitory potential of the examined essential oil on the formation of pathogenic bacteria biofilms is the work's central part (all excellently done). Therefore, the authors should consider slightly expanding the paper's introductory part regarding the biofilm matrix's structure. For example, at least add (e.g., “such as...”) typical polysaccharides and proteins in the biofilm composition (sentence on lines 48-50).
Small suggestions: (a) if something is (essentially) different, that difference must be significant. Therefore, if the word significant is used (as in the abstract, line 21 or line 115), then it would be better to state at what level that effect is statistically significant (p-value) or say that a particular difference exists (without “significantly”); (b) I think it is better to use the construction “biochemical processes” (line 157) than “biological processes” (the word biological is more associated with statics (structure), and biochemical with dynamics (activity)); (c) please replace (line 450) Tris-Cl with Tris-HCl; (d) the font size in figures 4–8 is (relatively) small compared to the rest of the text; consider increasing it; (e) consider also writing all Latin expressions in italics (via and vs.); (f) correct minor typos (like in lines 158, 202, 433, Table 1: name of compound No. 11); (g) I would remove “da Silva et al” from the line 239 (unnecessarily).
Comments on the Quality of English LanguageNo objections to the quality of the English language.
Reviewer 2 Report
Comments and Suggestions for Authors
Reviewer’s comments
The manuscript entitled “Transcriptional Profiling of the Effect of Coleus amboinicus L. Essential Oil against Salmonella Typhimurium Biofilm Formation” by Arpron et al., is a very critical information especially in current day and age where resistance to antibiotics is at its highest. The authors’ knowledge of the subject is obvious, and the manuscript is well written. Their work demonstrated that EO-CA has antibacterial properties against S. Typhimurium and blocked the formation of biofilm. I recommend the recommend the publication of this work with some minor modifications.
Minor Modifications.
1. Line 37-39: “Food is the source for most of these illnesses. Several common serovars, such as Salmonella enteritica serovar Typhimurium and S. Enteritidis, are the main species responsible for causing human infections” I suggest the addition of these two references here.
Gildea, L., Ayariga, J.A., Xu, J., Villafane, R., Robertson, B.K., Samuel-Foo, M. and Ajayi, O.S., 2022. Cannabis sativa CBD extract exhibits synergy with broad-spectrum antibiotics against Salmonella enterica subsp. Enterica serovar Typhimurium. Microorganisms, 10(12), p.2360.
Gildea, L., Ayariga, J.A., Ajayi, O.S., Xu, J., Villafane, R. and Samuel-Foo, M., 2022. Cannabis sativa CBD Extract Shows Promising Antibacterial Activity against Salmonella typhimurium and S. newington. Molecules, 27(9), p.2669.
2. Line 108-110: “The minimum inhibitory concentration (MIC) of EO-CA for S. Typhimurium was determined by the microbroth dilution was 1024 µg/mL and minimum bactericidal concentration (MBC) was 1024 µg/mL.” Revise this sentence to enhance flow of reading.
3. Line 196-198: Figure 7. (GO term enrichment analysis of the significant transcript levels...) show be stretched especially vertically to enhance clarity of the figure.
4. Line 199-202: “Figure 8. The most abundant pathways for a total of 104 DEGs.” The Figure should be enlarged to enhance figure quality, the pixel quality too should be enhanced.
5. Although the conclusion is succinct, the brevity of it leaves much out, please expand the conclusion two or three sentences further. The ability of EO-CA to inhibit S. Typhimurium at micromolar concentrations needs to be highlighted. There should be a statement connecting the transcriptomics analysis to the qPCR data, indicating the agreement between the two.
Reviewer 3 Report
Comments and Suggestions for Authors
The article titled "Transcriptional Profiling of the Effect of Coleus amboinicus L. Essential Oil against Salmonella Typhimurium Biofilm Formation." I must commend the authors for their comprehensive research, meticulous experimentation, and the valuable insights offered in this study.
· I am thoroughly impressed with the overall quality of the research and presentation. The study addresses an important issue related to Salmonella biofilm formation, which is of significant concern to public health and food safety.
· The potential of Coleus amboinicus L. essential oil (EO-CA) to inhibit biofilm formation and its transcriptional impact on S. Typhimurium is a significant contribution to the field.
· The article is well-organized and clearly presented, making it accessible to a wide range of readers. The logical flow from introduction to results and discussion helps readers follow the research process and its implications.
· I recommend the publication of this article and believe that it will greatly benefit researchers, and practitioners in the areas of microbiology, antimicrobial research, and food safety.
Corrections required:
The organism's name is not consistently italicized in some sections. Please make the necessary corrections to ensure consistent formatting.
Reviewer 4 Report
Comments and Suggestions for Authors
Despite a few number of observations, this manuscriptis well written.
General
Lines 27, 33, 35, 42, 69, 243, 262, 283, 350, 532: put Salmonella in italics
Results
Tables 1, 2 and 3: Remove internal lines.
Lines 108-109: check for grammatical errors
Lines 164-165: check for grammatical errors
Fig.3: How do you explain the fact 8x MIC had less postbiofilm inhibitory effect than 4x MIC for S. Typhimurium?
Discussion
Line 245: Write ‘‘biofilm development’’ instead of ‘‘development biofilm’’
Lines 264-266: check for grammatical errors
Line 331: Remove comma before ‘in’’.
Comments on the Quality of English LanguageThe following sentences must be rephrased for grammatical errors:
Lines 108-109: check for grammatical errors
Lines 164-165: check for grammatical errors
Line 245: Write ‘‘biofilm development’’ instead of ‘‘development biofilm’’
Lines 264-266: check for grammatical errors
Line 331: Remove comma before ‘in’’.
